# Evaluation of Postmortem Inspection Procedures to Diagnose Bovine Tuberculosis at Debre Birhan Municipal Abattoir

**DOI:** 10.3390/ani11092620

**Published:** 2021-09-07

**Authors:** Fanos Tadesse Woldemariyam, Tibebeu Markos, Dereje Shegu, Kassa Demissie Abdi, Jan Paeshuyse

**Affiliations:** 1College of Veterinary Medicine and Agriculture, Addis Ababa University, Bishoftu P.O. Box 34, Ethiopia; wfanostadessa@yahoo.com (F.T.W.); markostibebu@gmail.com (T.M.); kassa.demissie@hotmail.com (K.D.A.); 2Laboratory of Host Pathogen Interaction, Department of Biosystems, Division of Animal and Human Health Engineering, KU Leuven, 3001 Leuven, Belgium; 3National Animal Health and Diagnostic and Investigation Center, Sebeta P.O. Box 04, Ethiopia; dshegu@yahoo.com; 4College of Agriculture and Natural Resource Sciences, Debre Birhan University, Debre Birhan P.O. Box 445, Ethiopia

**Keywords:** abattoir, bovine tuberculosis, lesion, meat inspection

## Abstract

**Simple Summary:**

Tuberculosis is transmitted from animal to human by consuming raw or under cooked meat and milk from infected animals. Careful meat inspection considering all parts of the carcass is very important. However, most abattoirs in Ethiopia are performing only routine meat inspection which does not examine the organs in detail. Failure to do so might result in undetected lesions. In this study, two methods (routine and detailed meat inspections) were compared to inspect carcasses at abattoir level. Our study clearly shows that the routine meat inspection method misses about 89% of the tuberculosis lesions identified by detailed meat inspection. Based on detailed meat inspection, overall 4.7% of lesions suggestive of bovine tuberculosis (bTB) were found in the carcass of cattle slaughtered at Debre Birhan municipality abattoir during the study period. However, the routine meat inspection only detected 0.5% of the carcasses examined as having Tuberclosis (TB) lesions. Anatomically, 66% of the lesions were found in the lungs and associated lymph nodes, 21% in lymph nodes of the head, and 13% in the lymph nodes of the gastrointestinal tract. Therefore, lesion distribution routine and detailed meat inspection and their sensitivity with respect to lesion detection was identified.

**Abstract:**

Routine meat inspection in the abattoir was used to examine carcass for subsequent approval for consumption. However, the chance of missing lesions results in approval of carcass and/or the offal with lesions of tuberculosis. A cross-sectional study was conducted at Debre Birhan Municipal abattoir from October 2016 to May 2017. Lesion prevalence estimation and two meat inspection procedures’ efficacy evaluation was attempted. The breeds of the animals inspected were zebu breeds. Routine abattoir meat inspection involves visual inspection, palpation and incision of intact organs such as the liver and kidneys, as well as inspection, palpation and incision of tracheobronchial, mediastinal and prescapular lymph nodes. The detailed meat inspection involves inspection of each of the carcass. In this case, the seven lobes of the two lungs, lymph nodes and organs were also thoroughly examined. The cut surfaces were examined under bright light sources for the presence of an abscess, cheesy mass, and tubercles in detail. The study involved and compared both routine and detailed meat inspections at the abattoir. Chi-square test of independence and odds ratio were used to see the association of lesion and different risk factors. Based on detailed meat inspection, the overall lesion prevalence of bovine tuberculosis in the carcass of cattle slaughtered at Debre Birhan municipality abattoir was found to be 4.7% but only 0.5% of the carcass examined had detectable bovine tuberculosis lesions when routine abattoir meat inspection alone was used. The majority of the lesions were distributed to the lungs and associated lymph nodes. There was a significant association (*p* < 0.05) in TB infection rate and body condition score. In conclusion, this study has clearly indicated the prevalence of bovine tuberculosis lesions in the abattoir that are missed by routine abattoir meat inspection. In addition, it showed low sensitivity of the routine meat inspection procedure used. Hence, our study warrants immediate attention to strengthen the current meat inspection practices at Debre Birhan public abattoir.

## 1. Introduction

Tuberculosis is a chronic granulomatous debilitating disease causing mortality, morbidity and economic losses. It remains a major global health problem and causes ill-health among millions of people each year [1,2]. The disease is caused by a number of *M**ycobacterium* species of which *Mycobacterium bovis* is known to cause bTB. Bovine tuberculosis is a characteristic contagious disease affecting domestic animals and humans with granulomatous nodule (tubercles) formation in the affected organs. The location of this type of lesion depends on the route of infection and the age category affected. The transmission of the diseases in calves is through ingestion which facilitates lesion development in the mesenteric lymph nodes. In older cattle, infection is usually via the respiratory tract with lesions in the lungs and associated lymph nodes [3].

Animals affected by bovine tuberculosis lose 10–25% of their productive efficiency. Direct losses due to the infection become evident by a decrease in 10–18% milk and 15% meat production [4]. In addition, 10–15% of human cases of tuberculosis were reported in countries where pasteurization of milk is rare [5,6,7,8]. Tuberculosis also has a significant public health importance [8]. The disease in humans is currently becoming increasingly prevalent in developing countries, especially in rural areas, as humans and animals are sharing the same microenvironment and dwelling premises. Furthermore, patients with acquired immunodeficiency syndrome (AIDS) are more susceptible to tuberculosis [9].

In the Ethiopian context, the tuberculin skin test and abattoir meat inspection are the most commonly used techniques for bTB surveillance. Despite that, Mycobacterium culture is considered as the gold standard for TB diagnosis. Mycobacterium culturing is, however, very expensive, time consuming and unsafe in poorly equipped laboratories [10,11]. Several researches have researched the prevalence of bovine TB at animal and herd level in different parts of Ethiopia. Alelign et al. (2019) [12] showed a 1.5% (7/476) and 7.4% (7/95) prevalence at animal and herd level, respectively. Similarly, Mekonnen et al. (2019) [13] reported, using a lesion cut off value of >2 mm, a 9% and 65.5% prevalence at animal and herd level, respectively.

Different researchers reported an abattoir lesion prevalence of 3.5–5.6% in Zebu and 3.5–50% in crossbreed cattle in Ethiopia [5,9,14,15]. These reports did not represent the status of the disease in Ethiopia because of inadequate disease surveillance and a lack of better diagnostic facilities [16]. This particular study was designed to use two procedures, i.e., routine meat inspection followed by detailed meat inspection. Hence, aiming to increase the efficiency of lesion discovery. The authors are convinced that there were few studies in this regard which used two procedures to inspect meat in Ethiopia. Therefore, practicing an additional procedure would increase the chance of lesion discovery. Furthermore, as a result of missing lesions at abattoir level, the information about the genotypic characteristics of *M. bovis* strains affecting the cattle population in Ethiopia is limited. Such information is critical to monitor transmission and spread of the disease among cattle [5].

The two important risk factors for bTB infection of humans in Ethiopia are the consumption of raw animal products and sharing the same shelter with animals. Bovine tuberculosis is expected to have a significant impact on public health as 95% of the Ethiopian farmers still keep Zebu cattle using the traditional animal husbandry system [17]. Thus, the present study was initiated to address the specific objectives of estimating the abattoir lesion prevalence of bTB and assessing the lesion distributions in different tissues and organs in animals slaughtered at Debre Birhan municipality abattoir.

## 2. Materials and Methods

### 2.1. Study Areas and Animals

The study was conducted at Debre Birhan town Municipal abattoir from October 2016 to May 2017. November, December, March and April are religious fasting months for the local orthodox Christian majority of the town of Debre Birhan. During these months no animals are slaughtered and meat consumption is prohibited. Hence, during these months no carcasses could be inspected. Therefore, the study took place in October, January, February and May. During every week of each of these months there were 4 days of sampling and during each sampling day 6 animals were inspected. In total, there were 64 sampling days with a total of 384 carcasses inspected. Cattle were slaughtered at the abattoir four times a week. Thirty-eight cattle were slaughtered on average per day during the study period. All of them were local Zebu breeds (*Bos indicus*). Antemortem and postmortem procedures (routine and detailed meat inspections) were performed. During antemortem inspection, animals were examined irrespective of their origin, breed and sex. However, tracing the origin of the slaughtered cattle revealed that they originated from *Shoarobit, Kotugebeya, Chacha, Debre Birhan* and *Haik* (Figure 1). The majority were male animals with exception of very few females. Both medium (118 heads of cattle) and good body (266 heads of cattle) condition animals were slaughtered. There were no poor body condition animals as they were all brought from small scale traditional feedlots. All cattle slaughtered during the study duration were from 3 to 8 years of age and categorized as adult animals. Besides, the TB and tuberculin skin test status history of all cattle herds were not known to the investigators. The municipal inspectors were veterinarians with special training on meat inspection. The investigators were also veterinarians by training.

### 2.2. Study Design and Methodology

A cross-sectional study design was employed to determine the lesion prevalence of bTB and distribution in organs. A systematic random sampling procedure was used to select cattle for the current study. The animals needed to be selected based on their antemortem history (from the owner) and inspection in the lairage (inspector’s observation). All animals were found apparently healthy. The present study included 368 adult male and 16 adult female cattle (n = 384 in total). Six cattle were inspected by employing both routine and detailed procedures during each visit day. The meat inspection was done four times a week for four months. The sample size calculation was based on 50% expected prevalence since there was no abattoir lesion-based study on bTB in Debre Birhan abattoir. The appropriate formula to calculate the sample size for a 95% confidence interval and 5% desired absolute precision is adopted by [18].
$$n=\frac{Z^{2}\times p_{\exp}(1-p_{\exp})\times 100}{d^{2}}$$
where n = required sample size, p_exp_ = expected prevalence, d = desired absolute precision (5%), and Z = multiplier of 95% CI (1.96).

### 2.3. Antemortem Inspection

An antemortem inspection was performed on cattle included in the sample size before they were slaughtered. They all ought to be apparently healthy. Animal level traits such as identification number, breed, age, sex, and body condition scores (BCS) were observed and recorded in a logbook. The body condition of each of the animal studied was scored during antemortem examination by using a guideline established for Zebu by Nicholson (1986) [19]. In the meantime, the age of the study animals was also determined by using the dental eruption and wearing patterns described by De-Lahunta (1986) [20]. Additionally, body temperature, pulse rate, respiratory rate, type of nasal discharge (if present), condition of regional lymph nodes, and visible mucous membranes were clinically examined and recorded for individual animals to be slaughtered in conjunction with the postmortem findings.

### 2.4. Postmortem Inspection

#### 2.4.1. Routine Postmortem Inspection (RPMI)

The routine meat inspection procedure employed the protocol issued by [21,22]. The procedure involved visual inspection, palpation and incision of intact organs such as the liver and kidneys, as well as inspection, palpation and incision of tracheobronchial, mediastinal and prescapular lymph nodes. Further examinations of other lymph nodes and organs, or body systems were considered whenever lesions were detected in one of these tissues. The whole carcass was condemned if miliary TB involving multiple lymph nodes were detected, while the whole organs (or their parts) were condemned if large tuberculosis lesions were found in their parenchyma or associated lymph nodes [16]. Findings from routine meat inspection were recorded on meat inspection report forms.

#### 2.4.2. Detailed Postmortem Inspection (DPMI)

The inspection of each of the carcass was undertaken in detail [22,23]. Particular emphasis was given during examination to certain organs and lymph nodes that were carefully inspected for the presence of suspected bTB lesions (Table 1). The seven lobes of the two lungs, lymph nodes and organs were also thoroughly examined. The cut surfaces were examined under bright light sources for the presence of an abscess, cheesy mass, and tubercles [24]. When gross lesions suggestive of bTB were found in any of the tissues, the whole carcass was classified as having lesion. The frequency of lesions in certain anatomical sites with TB like lesions was recorded. These organs with detectable lesions were condemned and disposed of appropriately.

### 2.5. Ethical Clearance

The current study was conducted on those cattle brought to be slaughtered at Debre Birhan municipal abattoir. Therefore, no ethical clearance was needed.

### 2.6. Data Collection and Analysis

The data collected and recorded during abattoir sampling were entered and stored in separate MSExcel (version:14.0.4734.1000) spread sheets, thoroughly screened for errors, coded, imported and analyzed in STATA version 13.0 and Epicalc 2000 for Windows. Descriptive statistics was used to summarize the frequency and prevalence data. Prevalence was calculated as the proportion of suspected lesion positive animals to the total number of animals inspected. The association of TB lesion with the predictor variables was analyzed by making use of chi-square (χ^2^) test of independence. Moreover, the strength of associations of TB lesion with potential risk factors was analyzed by using the odds ratio. The 95% CI was used and all statistical associations were considered significant when *p* < 0.05 [25].

## 3. Results

### 3.1. Antemortem Inspection

Upon antemortem inspection in the lairage, all six animals per day selected for inspection were found physically fit for slaughter. In total, 384 animals were examined in the study period but none of them showed any noticeable physical abnormalities.

### 3.2. Lesion Prevalence of Bovine Tuberculosis

The overall lesion prevalence of bTB in cattle slaughtered at Debre Birhan municipality abattoir during the study period was 4.7% (18/384) at 95% CI [2.89, 7.46] based on detailed meat inspection as indicated in Table 2, Table 3 and Table 4. However, only 0.5% (2/384) at 95% CI [0.09, 2.08] of cattle was found to have detectable TB like lesions by the routine meat inspection as shown in Table 2.

The sensitivity of routine meat inspection at Debre Birhan abattoir was 11%. From the result of this study, the probability of missing an animal with TB suspected lesion during routine abattoir meat inspection was estimated to be 89%. However, the specificity of detailed meat inspection was found to be high which could maximize the efficiency of lesion detection (Table 2).

On the other hand, the association of age and sex in detailed meat inspection was statistically insignificant (*p* > 0.05). However, there was a statistically significant association (*p* < 0.05) between body condition of the slaughtered cattle and the occurrence of TB like lesions indicated in Table 4.

### 3.3. Gross Pathology

The most common gross pathological changes seen in the affected organs and/or lymph nodes were the presence of circumscribed yellowish white lesions of various sizes and numbers in tracheobronchial lymph nodes. Miliary lesions were observed in the lungs and liver in 11% (2/18) of bTB affected carcasses. Large encapsulated nodules containing yellowish white exudate were observed in tracheobronchial, submandibular and mesenteric lymph nodes. There were also active lesions characterized by reddish to black demarcated areas in the lungs, bronchial (caseous), tracheobronchial, mediastinal, hepatic and mesenteric lymph nodes. Calcified and necrotized lesions were also seen in the lungs and submandibular lymph nodes. Some of the gross pathological changes observed during abattoir inspection were as depicted in Figure 2 and Table 5.

The distributions of lesions in different tissues of cattle observed in seven organs and/or lymph nodes were containing tuberculous lesions. About 65.8% (25/38) of the lesions were observed in the lungs and associated lymph nodes followed by 21.1% (8/38) and 13.2% (5/38) in cervical and gastrointestinal lymph nodes, respectively, indicated in Table 5.

## 4. Discussion

The present study was conducted at Debre Birhan municipal abattoir that provides slaughter and inspection service to the town. The animals were adult feedlot cattle brought from five different places for slaughter and sold at butcher houses of the town. The butchers were serving the residents and visitors raw or cooked meat. Both routine and detailed meat inspection techniques were employed. Hence, the overall lesion prevalence of 0.5% and 4.7% was obtained at the animal level based on routine and detailed meat inspection techniques, respectively.

Prevalence reports of bTB from Addis Ababa Abattoirs Enterprise, Hosanna municipal abattoir and Yabello municipal abattoir indicated 3.0, 4.5 and 4.2% lesion prevalence [14,16,26], respectively, by detailed meat inspection comparable to our finding. However, the present finding was slightly lower than 5.8%, 6.4%, 6.8%, reported from different abattoir in Ethiopia [27,28,29]. Much higher lesion prevalence was also reported from Butajira, Addis Ababa, Afar, Adama and Gambella municipal Abattoir with 11.5, 10.1%, 11.0%, 24.7% and 13.2%, respectively [14,29,30,31,32]. In addition, our finding was also lower than the incidence reported for rural Tanzania, which is a 20% lesion prevalence [33]. These findings could indicate the endemic nature of the disease and high infection rates prevailing in the general population of slaughtered cattle in the study areas.

Our finding implies that a large proportion, i.e., 89%, of tuberculosis infected carcasses passed undetected and the meat is approved for human consumption. Previous studies have also indicated probabilities of 85%, 84% and 70% [24,26,34] of missing an animal with TB lesion. Therefore, detailed meat inspection can be considered as a better procedure to detect TB lesions.

The most probable explanation for the failure of standard meat inspection to correctly detect tuberculosis infection could be the manner of examination [24]. It was noticed that in standard meat inspection procedure only a few sites (organs) are often inspected at a glance due to the work burden of inspecting a large number of animals each day as described by Corner (1994) [24]. In addition, lack of interest of inspectors as well as absence of institutional will to incorporate or enforce both methods of examinations as a standard meat inspection workflow. These two might be possible reasons for the inefficiency of the service.

Gross tuberculosis like lesions in this study were observed most frequently in the lungs and the associated lymph nodes (65.8%) followed by lymph nodes of the head (21.1%) and lymph nodes of the gastrointestinal tract (13.2%). This finding is similar to reports from previous studies done in Ethiopia where 70% TB lesions were reported in lungs and associated lymph nodes [35]. This is also in agreement with Corner (1994) [24] that reported up to 95% of visible TB lesions in the lung and associated lymph nodes. The result, therefore, indicated that the primary route of infection was through the respiratory route which can also spread to other parts of the body as described previously [4]. Therefore, during postmortem examination, lungs and associated lymph nodes should be the focus of the inspection. However, the presence of lesions in mesenteric lymph nodes also indicates that the additional infection could occur through ingestion [4].

In all age groups (3–5 and 6–8 years) the lesions were identified at a lower rate. This is in agreement with previous reports [26,36] that recorded the presence of lesions in all age groups. All age groups slaughtered can develop the lesions once exposed for continuous infection over an extended duration. On the other hand, there was a statistically significant association (*p* < 0.05) between lesion presence and BsS of slaughtered cattle. The infection occurred in medium (13.6%) as compared to good body condition animals (0.7%) which is in agreement with the findings by Biratu et al. (2014) [37]. This could indicate the wasting nature of the disease and also that animals with good BCS have relatively good immunological response to the infectious agent compared to animals with medium BCS [4].

## 5. Conclusions

The present study has once again confirmed the existence of bovine tuberculosis infected animals slaughtered at Debre Birhan municipal abattoir. Detailed postmortem inspection was found to be better than routine postmortem inspection. The study also revealed a high proportion of tuberculous lesions in the lungs and lymph nodes of the thoracic cavity indicating the respiratory system to be the predominant route of infection.

Based on the above concluding remark, the following recommendations are made: (i) training of the meat inspectors working at Debre Birhan municipal abattoir so as to safeguard more people or to protect as many people as possible; (ii) an in-depth epidemiological study focusing on culturing, isolation and identification of the agent as well as molecular characterization of the strains cycling in the study area; (iii) public awareness campaign should be done about zoonotic importance of bovine TB in humans via the consumption of raw or undercooked animal products.

## Figures and Tables

**Figure 1 animals-11-02620-f001:**
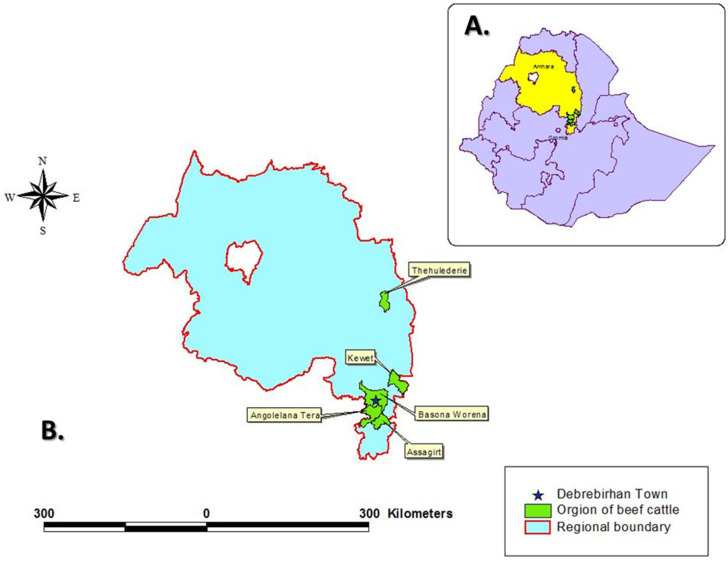
Geographical location of the study sites and beef cattle sources. (**A**) Map of Ethiopia with the Regional State of Amhara, in which this study was conducted, highlighted in yellow. (**B**) Map of the Regional State of Amhara with the town of Debrebirhan (**★**) and the sampling areas of Thehulederie, Kewet, Angolelana Tera, Assagirt and Basona Worena highlighted in green.

**Figure 2 animals-11-02620-f002:**
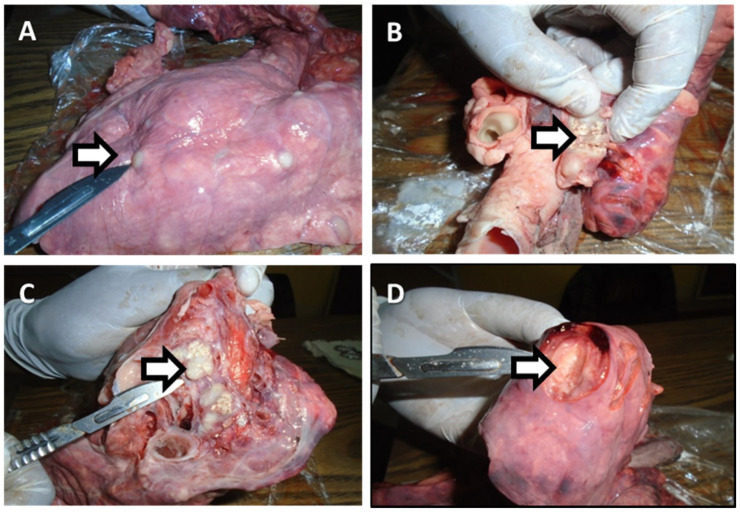
Typical TB lesions (indicated by an ⇨) of cattle slaughtered at Debre Birhan town municipal abattoir: (**A**) nodules in the lungs of an ox; (**B**) caseation of bronchial lymph node in the lungs of an ox; (**C**) calcified mediastinal lymph node in the lungs of cow; (**D**) white spots in the lung tissue of an ox.

**Table 1 animals-11-02620-t001:** Anatomical location of lymph nodes in different parts of the body.

Organ	Lobes/Parts
Lung	Left apical, left cardiac, left diaphragmatic, right apical, right cardiac, right diaphragmatic and right accessory lobes
Lymph node	Parotid, submaxillary, submandibular, lateral and medial retropharyngeal, tracheobroncheal, cranial and caudal mediastinal, hepatic, mesenteric, iliac, precrural, presacpular, supramammary, inguinal, apical, and ischiatic
Organ	Liver, kidneys, mammary glands and intestines

**Table 2 animals-11-02620-t002:** Contingency table (2 × 2) to compare routine and detailed abattoir inspections.

RPMI	DPMI		
	Positive	Negative	Total
Positive	2	0	2
Negative	16	366	382
Total	18	366	384

RPMI = routine postmortem inspection; DPMI = detailed postmortem inspection; sensitivity = 11% ([1.91, 35.95]; specificity = 100% [98.71, 99.98])

**Table 3 animals-11-02620-t003:** Lesion description based on routine and detailed postmortem inspection.

Postmortem Findings	Observation	Percent (%)
Granulomatous lesion	18	4.7
No lesion	366	95.3
Total	384	100

**Table 4 animals-11-02620-t004:** Summary of the risk factors in association to postmortem meat inspections.

Variables	n	RPMI			DPMI				
		Positives	Prevalence	χ^2^	Positives	Prevalence	OR	χ^2^	*p*-Value
			% [95% CI]			% [95% CI]	[95% CI]		
Age	384	2	0.52 [0.09, 2.08]	0	18	4.7 [2.89, 7.46]	2.79 [0.79, 9.81]	3.1	0.11
3–5 years	134	0	0		3	2.2 [0.56, 6.85]			
6–8 years	250	2	0.8 [0.14, 3.17]		15	6.0 [3.52, 9.90]			
Sex	384	2	0.52 [0.09, 2.08]	0	18	4.7 [2.89, 7.46]	0.32 [0.07, 1.53]	1.63	0.151
Female	16	0	0		2	12.5 [2.20, 39.59]			
Male	368	2	0.54 [0.09, 2.16]		16	4.35 [2.59, 7.11]			
BCS	384	2	0.52 [0.09, 2.08]	0	18	4.7 [2.89, 7.46]	0.05 [0.01, 0.21]	28.1	0
Medium	118	2	1.7 [0.30, 6.61]		16	13.6 [8.22, 21.43]			
Good	266	0	0		2	0.75 [0.13, 2.98]			

BSC = body condition score; OR = odds ratio; CI= confidence interval, DPMI = detailed postmortem inspection.

**Table 5 animals-11-02620-t005:** Distribution of lesions based on anatomical sites.

S/No.	Tissue and Lymph Nodes	RPMI	DPMI	Total no. of Lesions	Proportion (%)
1	Lungs	1	14	15	39.47
2	Mediastinal		3	3	7.90
3	Retropharyngeal	1	5	6	15.79
4	Submandibular		2	2	5.26
5	Mesenteric		3	3	7.90
6	Bronchial		7	7	18.42
7	Hepatic		2	2	5.26
Total				38	100

RPMI = routine postmortem inspection; DPMI = detailed postmortem inspection; S/No: sample number; no. number.

## Data Availability

The data presented in this study are available on request from the corresponding author. The data are not publicly available due to privacy of the abattoir customers.

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
