# Peer review of "Evaluation of Postmortem Inspection Procedures to Diagnose Bovine Tuberculosis at Debre Birhan Municipal Abattoir"

_animals, 2021, doi:10.3390/ani11092620_

Round 1

Reviewer 1 Report

Thank you for letting me review this interesting article, which highlights the importance of good postmortem practices at slaughterhouses to detect bovine tuberculosis lesions. I think that the manuscript has improved considerably, however, still it has some major flaws that need to be addressed. 

Therefore, I recommend that a revision. I explain my concerns in more detail below, and I ask that the authors to address them in the manuscript.

Abstract
Abstract describes the main problem and the objective of the study (estimate prevalence of lesion and compare two inspection procedures: routine and detailed meat inspection), and some conclusions. However, the mentioned conclusion "this study has clearly indicated the prevalence of bTB lesion in the abattoir" does not provided any information or conclusion. Is the sentence incomplete?

L 48-49: These tests are used to check for associations. What the authors mean with "analyze the data" exactly? Could you please review the sentence?

L 53-54: Wat are "epidemiological safety measures"? Please clarify the sentence.

Materials and methods
Still it is not clear how the authors decided to study 384 animals, and what were the selection criteria, as they did not include all animals slaughtered in that abattoir. If inspection carried out 4 times (or days?) per week, with 6 animals inspected per day equals 24 animals per week. If this study lasted 8 months = 32 weeks, it should be 768 animals inspected.

L 122- 123 How an antemortem procedure was performed in a routine and detailed meat inspection? Later in L 155 it is mentioned that "cattle.. were clinically examined". Please clarify.

L 135: It is a repetition of line 121-122. Should you delete any of them.

L 142: How can be unfortunate that animals did not have respiratory illness? Of course, it was not appropriate for the study, but please, rewrite the sentence.

L 167-169: Please adapt the cited references to the sentence.

L 179-186: What is the difference with RPMI? How the examination in the DPMI took place? Where the animals with detected gross lesions condemned as in RPMI?

Results
L 203-205: It is a repetition from previous section. Please remove or rewrite.

Table 3: In previous tables it is mentioned that 18 samples were positive to DPMI, but table 3 only reports results from 16 samples.

Table 4: Could authors review this table? It seems that columns 3-5 are for RPMI and  6-10 are for DPMI, but this piece of information is missed.

L 217-218: I agree that culturing Mycobacterium bovis is the gold standard to detect the bacteria, but how can it be for detection of lesions at abattoirs?  Please could the authors provide an explanation?

L 220-221: Do the authors carry out a multivariate analysis to evaluate collinearity as mentioned? In such case, please provide the results. If not, please review the sentence.

Discussion
L 269-270: the study detected a much lower sensitivity of 11%, compared to what? The sentence is incomplete.

L 275-276: Depending on the abattoir, region, country, etc. 38 animals as average per day, is not consider a large number. Could authors comment on this?

L 277-279: Authors explained the low percentage of detection of lesions because the personnel lack adequate training. However, L 132-135 stated that inspectors are veterinarians with special training on meat inspection. Please clarify.

L 287-288: Prevalence are not significant or not, they are value/percentage. What it is statistically significant or not is the difference in prevalence between groups. Please correct the sentence.

L 290-291: Do the authors mean a statistically significant association between prevalence of detection of lesions in the abattoir and BCS of slaughtered cattle? Please correct.

L 296-301 does not discuss results from the present study but reports what other have stated. Please discuss with

your own results of distribution of TB lesions.

General comments

Be consistent with the terminology, for example bTB or bovine tuberculosis (L 77, 91, 111).

Change the numbering of the subsections 2.6.1 and 2.6.2 when it should read 2.4.1 and 2.4.2

Name the tables in order of appearance in the text.

Author Response

Ms.Ref. No.: animals-1319917

Dear Editor

Please find herewith our point by point correction of the valuable comments of reviewer 1 regarding manuscript entitled “Evaluation of Postmortem Inspection Procedures to Diagnose Bovine Tuberculosis at Debre Birhan Municipal Abattoir; authored by Fanos et al. We revised our manuscript according to the suggestions of the Reviewers. We also included an annotated version of the manuscript with highlighted changes. We responded as follows to the comments:

Sincerely

Jan Paeshuyse

Reviewer 1

Thank you for letting me review this interesting article, which highlights the importance of good postmortem practices at slaughterhouses to detect bovine tuberculosis lesions. I think that the manuscript has improved considerably, however, still it has some major flaws that need to be addressed. 

Therefore, I recommend that a revision. I explain my concerns in more detail below, and I ask that the authors to address them in the manuscript.

Q1: Abstract
Abstract describes the main problem and the objective of the study (estimate prevalence of lesion and compare two inspection procedures: routine and detailed meat inspection), and some conclusions. However, the mentioned conclusion "this study has clearly indicated the prevalence of bTB lesion in the abattoir" does not provided any information or conclusion. Is the sentence incomplete?

A1:As per the reviewers suggestion the abstract was modified as follows

Routine meat inspection in the abattoir was used to examine carcass for subsequent approval for consumption. But the chance of missing lesions results in approval of carcass and/or the offal with lesions of tuberculosis. A cross-sectional study was conducted at Debre Birhan Municipal abattoir from October 2016 to May 2017. Lesion prevalence estimation and two meat inspection procedures efficacy evaluation was aimed. The breeds of the animals inspected were zebu breeds. Routine abattoir meat inspection involves visual inspection, palpation and incision of intact organs such as the liver and kidneys, as well as inspection, palpation and incision of tracheobronchial, mediastinal and prescapular lymph nodes.  The detailed meat inspection involves inspection of each of the carcass was in detail. The study involved and compared both routine and detailed meat inspections at the abattoir. Chi-square test of independence, and odds ratio were used to see the association of lesion and different risk factors. Based on detailed meat inspection, the overall lesion prevalence of bovine tuberculosis in the carcass of cattle slaughtered at Debre Birhan municipality abattoir was found to be 4.7% but only 0.5% of the carcass examined had detectable bovine tuberculosis lesions when routine abattoir meat inspection alone was used.  The majorities of the lesions were distributed to the lungs and associated lymph nodes. There was a significant association (P <0.05) in TB infection rate and body condition score. In conclusion, this study has clearly indicated the prevalence of bovine tuberculosis lesions in the abattoir that are missed by routine abattoir meat inspection. In addition it showed low sensitivity of the routine meat inspection procedure used. Hence, our study warrants immediate attention to strengthen the current meat inspection practices at Debre Birhan public abattoir.

Q2: L 48-49: These tests are used to check for associations. What the authors mean with "analyze the data" exactly? Could you please review the sentence?

A2:As per the reviewer’s suggestion the phrase "analyze the data"   is replaced by ‘to see the association of lesion and different risk factors’. As shown below

 Chi-square test of independence, and odds ratio were used to see the association of lesion and different risk factors.

Q3:L 53-54: Wat are "epidemiological safety measures"? Please clarify the sentence.

 A3:  Removed from the current version of the abstract

Q4:Materials and methods
Still it is not clear how the authors decided to study 384 animals, and what were the selection criteria, as they did not include all animals slaughtered in that abattoir. If inspection carried out 4 times (or days?) per week, with 6 animals inspected per day equals 24 animals per week. If this study lasted 8 months = 32 weeks, it should be 768 animals inspected.

A4:Dear reviewer you are right if the study was done for eight months it should be 768. From mid-November to end of December, the whole March and April months were fasting periods as the area was predominantly Christian society. So these weeks were not included in our study. But what we did was using  Michael Thrusfield, 2005 formula prevalence sampling for prevalence was calculated. Accordingly we used 50% of the animals coming to the abattoir through the entire period of the study had bTB lesion. By this calculation we set our sample size to 384.This was inspected when the abattoir is functional. As I mentioned above it was not functional from mid-November to end of December and again from March to end of April. These months are the fasting period of the dominant Christian society in the area. So the study period was from October to march but mainly 15 weeks visit was performed

4x6x15=360

Q5:L 122- 123  How an antemortem procedure was performed in a routine and detailed meat inspection? Later in L 155 it is mentioned that "cattle.. were clinically examined". Please clarify.

A5:As per the reviewers suggestion the statement “antemortem procedure was performed in a routine and detailed meat inspection” is replaced by ‘Antemortem and postmortem procedures (routine and detailed meat inspections) were performed’.

"cattle.. were clinically examined" is replaced by  An ante mortem inspection was performed on cattle included in the sample size before they were slaughtered

Q6:L 135: It is a repetition of line 121-122. Should you delete any of them?

A6:As per the reviewers suggestion ‘All the animals slaughtered were zebu breeds’. Deleted.

Q7:L 142: How can be unfortunate that animals did not have respiratory illness? Of course, it was not appropriate for the study, but please, rewrite the sentence.

A7:As per the review suggestion the sentence ‘Unfortunately no animals did not have respiratory illness’ replaced by ‘All animals were found apparently healthy’

Q8:L 167-169: Please adapt the cited references to the sentence.

A8:As per the reviewers suggestion the reference the sentence was adapted to the reference as :-‘ The routine meat inspection procedure employed the protocol issued by (Hailemariam, 1975) and (Ameni et al., 2007)’.

Q9:L 179-186: What is the difference with RPMI? How the examination in the DPMI took place? Where the animals with detected gross lesions condemned as in RPMI?

A9:This is the procedure of DPMI: Particular emphasis was given during examination to certain organs and lymph nodes that were carefully inspected for the presence of suspected bTB lesions (Table 6). The seven lobes of the two lungs, lymph nodes and organs were also thoroughly examined. The cut surfaces were examined under a bright light sources for the presence of an abscess, cheesy mass, and tubercles (Corner, 1994). When gross lesions suggestive of bTB were found in any of the tissues, the animal was classified as having lesions. The frequency of anatomical sites where TB like lesions detected was recorded.

Yes, these organs with detectable lesions were condemned and disposed appropriately.

Results

Q11:L 203-205: It is a repetition from previous section. Please remove or rewrite.

A11: Thanks for the suggestion but this part is the result of ante mortem inspection which is different from the previous one.

Q12:Table 3: In previous tables it is mentioned that 18 samples were positive to DPMI, but table 3 only reports results from 16 samples.

A12:As per the suggestions the table was corrected and the numbers were set 18

Q13:Table 4: Could authors review this table? It seems that columns 3-5 are for RPMI and  6-10 are for DPMI, but this piece of information is missed.

A13:As per the reviewers suggestion table four was  reviewed and column 3-5 are for RPMI AND 6-10 are DPMI.

Q14:L 217-218: I agree that culturing Mycobacterium bovis is the gold standard to detect the bacteria, but how can it be for detection of lesions at abattoirs?  Please could the authors provide an explanation?

A14: Hundred percent detection of TB lesion is expected from the gold standard method of culturing Mycobacterium bovis.

This statement was written here just to make a comparison with the lesion detection capacity of DPMI. Which, by far is lower than culturing but it is best to apply in areas were laboratory facilities are scarce. 

Q15:L 220-221: Do the authors carry out a multivariate analysis to evaluate collinearity as mentioned? In such case, please provide the results. If not, please review the sentence.

A15:This statement was deleted :The association of age, Body Condition Score and sex to the occurrence of tuberculosis like lesions in routine postmortem inspection was not appreciated due to collinearity.

Discussion
Q16:L 269-270: the study detected a much lower sensitivity of 11%, compared to what? The sentence is incomplete.

A16:This is 89% miss of the lesion ,of course it is a redundancy  of  the below sentence and it is removed

Q17:L 275-276: Depending on the abattoir, region, country, etc. 38 animals as average per day, is not consider a large number. Could authors comment on this?

A17: Well, Debre berehan is a big town with a number of butcher houses. The population as well as the big hotels are also served by this abattoir. 38 animals will not be large number.

Q18:L 277-279: Authors explained the low percentage of detection of lesions because the personnel lack adequate training. However, L 132-135 stated that inspectors are veterinarians with special training on meat inspection. Please clarify.

A18:This is to mean use of routine postmortem meat inspection protocol is not designed in such a way that you can cover the detail of each organ and offal. Simply you do major lymph nodes .Even if they are trained to do meat inspection the practice they are making is only RPMI which is less sensitive. In terms of their training it was given once in their carrier and that might be lost or forgotten during their stay. It needs continuous professional development.

Q19:L 287-288: Prevalence are not significant or not, they are value/percentage. What it is statistically significant or not is the difference in prevalence between groups. Please correct the sentence.

A19:As per the reviewers suggestion the paragraph is re-written as follows:- In all age groups the lesion was identified rate This is in agreement with previous reports (Bekele, 2011; Teklu et al., 2004) that recorded the presence of lesion in all age groups. All age groups slaughtered can develop the lesions once exposed for continuous infection over an extended duration. On the other hand, there was a statistically significant association (P < 0.05) between abattoir lesion presence and BCS of slaughtered cattle. L 290-291: Do the authors mean a statistically significant association between prevalence of detection of lesions in the abattoir and BCS of slaughtered cattle? Please correct.

Q20:L 296-301 does not discuss results from the present study but reports what other have stated. Please discuss with your own results of distribution of TB lesions.

A20:As per the suggestion the paragraph was re-written as follows: In this study 65.8% (25/38) of the lesions were observed in the lungs and associated lymph nodes followed by 21.1% (8/38) and 13.2% (5/38) in cervical and gastrointestinal lymph nodes, respectively. This is in agreement with Corner, (1994) that reported up to 95% of visible TB lesions in the lung and associated lymph nodes. This finding indicates that inhalation might be the principal route of TB infection in cattle. Therefore, during postmortem examination, lungs and associated lymph nodes should be the focus of the inspection. However, the presence of lesions in mesenteric lymph nodes also indicates that the additional infection could occur through ingestion (Radostits et al.,2007)

Q21:General comments

Be consistent with the terminology, for example bTB or bovine tuberculosis (L 77, 91, 111).

A20:As per the reviewers suggestion it was corrected as ‘Bovine tuberculosis (bTB)’ was written

Q22:Change the numbering of the subsections 2.6.1 and 2.6.2 when it should read 2.4.1 and 2.4.2

A22:As per the suggestion it is corrected to 2.4.1 and 2.4.2

Q23:Name the tables in order of appearance in the text.

A23:As per the suggestions the tables are reordered and referred according to their appearance

Reviewer 2 Report

General comments

The manuscript has been considerably improved by making the recommended revisions, in particular the description of the various paragraphs and of the tables. The sections Materials and Methods, Results and Discussion are now more complete and clearer. The conclusions are well introduced, presented, and adequately discussed. The sentences have been improved but some should be rewritten because still unclear and difficult to understand. References are up to date, but not written in the correct format both in the text and at the end of the manuscript; the authors must necessarily adapt the text according to the guidelines of the journal. The Discussion has been properly reorganized. In my view, this manuscript is now suitable for publication in Animals after other minor Revisions in order to improve the document.

Specific comments

The simple summary and the abstract are too long and almost a repetition of each other; they must be necessarily synthesized. For ease, I copy and paste the explanations of these two sections that are presented in the guidelines of Animals:

Simple Summary: The simple summary consists of no more than 200 words in one paragraph and contains a clear statement of the problem addressed, the aims and objectives, pertinent results, conclusions from the study and how they will be valuable to society. This should be written for a lay audience, i.e., no technical terms without explanations. No references are cited and no abbreviations.

Abstract: The abstract should be a total of about 200 words maximum. The abstract should be a single paragraph and should follow the style of structured abstracts, but without headings: 1) Background: Place the question addressed in a broad context and highlight the purpose of the study; 2) Methods: Briefly describe the main methods or treatments applied. Include any relevant preregistration numbers, and species and strains of any animals used. 3) Results: Summarize the article's main findings; and 4) Conclusion: Indicate the main conclusions or interpretations. The abstract should be an objective representation of the article: it must not contain results which are not presented and substantiated in the main text and should not exaggerate the main conclusions.

Keywords: Please modify as follows: Abattoir; Bovine tuberculosis; Lesion; Meat Inspection.

  1. Materials and Methods

Line 126. Figure 1.

Line 135. “All the animals slaughtered were zebu breeds”. This is a redundant information, please delete it.

Line 143. Replace “comprises” with a more scientific term and write it in the past tense, “included” for example.

Lines 185-186. “The frequency of anatomical sites where TB like lesions detected was recorded”. Please re-write this sentence because it is unclear.

  1. Results

Lines 216-217. “However, the specificity of detailed postmortem inspection was found to be high which could reasonably detect TB-like lesions indicated in table 3”. Please re-write the entire sentence because it is unclear.

  1. Discussion

Lines 246-248. “The present study was conducted at Debre Birhan municipal abattoir on adult feedlot cattle brought from five different places for slaughter and sold at butcher houses found at Debre Birehan. This can be consumed by the residents and visitors as raw or cooked meat”. Please reorganize these first sentences of the introduction.

References: Please write the references according to the guidelines of the journal. They must all be changed and not written in alphabetically order.

Table 3. +Ve: positive.

Table 6. The table is not completely visible, please adapt it to the page.

Figure 1. Orgion of beef cattle?

Author Response

Ms.Ref.No.: animals-1319917

Dear Editor

Please find herewith our point by point correction of the valuable comments of reviewer 2 regarding manuscript entitled “Evaluation of Postmortem Inspection Procedures to Diagnose Bovine Tuberculosis at Debre Birhan Municipal Abattoir; authored by Fanos et al. We revised our manuscript according to the suggestions of the Reviewers. We also included an annotated version of the manuscript with highlighted changes. We responded as follows to the comments:

Sincerely

Jan Paeshuyse

Reviewer 2

General comments

The manuscript has been considerably improved by making the recommended revisions, in particular the description of the various paragraphs and of the tables. The sections Materials and Methods, Results and Discussion are now more complete and clearer. The conclusions are well introduced, presented, and adequately discussed. The sentences have been improved but some should be rewritten because still unclear and difficult to understand. References are up to date, but not written in the correct format both in the text and at the end of the manuscript; the authors must necessarily adapt the text according to the guidelines of the journal. The Discussion has been properly reorganized. In my view, this manuscript is now suitable for publication in Animals after other minor Revisions in order to improve the document.

Specific comments

The simple summary and the abstract are too long and almost a repetition of each other; they must be necessarily synthesized. For ease, I copy and paste the explanations of these two sections that are presented in the guidelines of Animals:

Q1:Simple Summary: The simple summary consists of no more than 200 words in one paragraph and contains a clear statement of the problem addressed, the aims and objectives, pertinent results, conclusions from the study and how they will be valuable to society. This should be written for a lay audience, i.e., no technical terms without explanations. No references are cited and no abbreviations.

A1:As per the reviewers suggestion the simple summary was corrected as follows

Tuberculosis is transmitted from animal to human by consuming raw or under cooked meat and milk from infected animals. Careful abattoir inspection considering all parts of the carcass is very important. But most abattoirs in Ethiopia are performing only routine meat inspection which does not examine the organs in detail. Failure to do so might result in lesions undetected. In this study two methods (routine and detailed meat inspections) were compared to inspect carcasses at abattoir level. Our study clearly shows that the routine meat inspection method misses about 90% of the tuberculosis lesion identified by detailed meat inspection. Based on detailed meat inspection, overall 4.7% of lesion prevalence suggestive of bovine tuberculosis was found in the carcass of cattle slaughtered at Debre Birhan municipality abattoir during the study period. However, the routine abattoir meat inspection only detected 0.5% of the carcasses examined as having TB lesions. Anatomically, 66% of the lesions were found in the lungs and associated lymph nodes, 21% in lymph nodes of the head, and 13% in the lymph nodes of the gastrointestinal tract. Therefore, lesion distribution, importance of combining routine and detailed meat inspection and their sensitivity with respect to lesion detection was identified.

Q2:Abstract: The abstract should be a total of about 200 words maximum. The abstract should be a single paragraph and should follow the style of structured abstracts, but without headings: 1) Background: Place the question addressed in a broad context and highlight the purpose of the study; 2) Methods: Briefly describe the main methods or treatments applied. Include any relevant preregistration numbers, and species and strains of any animals used. 3) Results: Summarize the article's main findings; and 4) Conclusion: Indicate the main conclusions or interpretations. The abstract should be an objective representation of the article: it must not contain results which are not presented and substantiated in the main text and should not exaggerate the main conclusions.

A2:As per the reviewers suggestion the abstract was modified as follows

Routine meat inspection in the abattoir was used to examine carcass for subsequent approval for consumption. But the chance of missing lesions results in approval of carcass and/or the offal with lesions of tuberculosis. A cross-sectional study was conducted at Debre Birhan Municipal abattoir from October 2016 to May 2017. Lesion prevalence estimation and two meat inspection procedures efficacy evaluation was aimed. The breeds of the animals inspected were zebu breeds. Routine abattoir meat inspection involves visual inspection, palpation and incision of intact organs such as the liver and kidneys, as well as inspection, palpation and incision of tracheobronchial, mediastinal and prescapular lymph nodes.  The detailed meat inspection involves inspection of each of the carcass was in detail. The study involved and compared both routine and detailed meat inspections at the abattoir. Chi-square test of independence, and odds ratio were used to analyze the data. Based on detailed meat inspection, the overall lesion prevalence of bovine tuberculosis in the carcass of cattle slaughtered at Debre Birhan municipality abattoir was found to be 4.7% but only 0.5% of the carcass examined had detectable bovine tuberculosis lesions when routine abattoir meat inspection alone was used.  The majorities of the lesions were distributed to the lungs and associated lymph nodes. There was a significant association (P <0.05) in TB infection rate and body condition score. In conclusion, this study has clearly indicated the prevalence of bovine tuberculosis lesion in the abattoir. In addition it showed low sensitivity of the routine meat inspection procedure used. Hence, our study warrants immediate attention to strengthen the current meat inspection practices at Debre Birhan public abattoir.

Q3:Keywords: Please modify as follows: Abattoir; Bovine tuberculosis; Lesion; Meat Inspection.

A3: As per the reviewers suggestion the Keywords are modified Abattoir; Bovine tuberculosis; lesion; meat inspection. To Abattoir; Bovine tuberculosis; Lesion; Meat Inspection.

  1. Materials and Methods

Q4:Line 126. Figure 1.

A4:As per the reviewers suggestion figure 1 was corrected to Figure 1

Q5:Line 135. “All the animals slaughtered were zebu breeds”. This is a redundant information, please delete it.

A5:As per the reviewers suggestion this statement is  deleted :  ‘All the animals slaughtered were zebu breeds’.

Q6: Line 143. Replace “comprises” with a more scientific term and write it in the past tense, “included” for example.

A6:As per the reviewers suggestion the word ‘comprises’ is replaced by’ included’

Q7:Lines 185-186. “The frequency of anatomical sites where TB like lesions detected was recorded”. Please re-write this sentence because it is unclear.

A7:As per the reviewers suggestion the sentence “The frequency of anatomical sites where TB like lesions detected was recorded” was re written to ‘The frequency of lesions in a certain anatomical sites with TB like lesions was recorded’

  1. Results

Q8: Lines 216-217. “However, the specificity of detailed postmortem inspection was found to be high which could reasonably detect TB-like lesions indicated in table 3”. Please re-write the entire sentence because it is unclear.

A8:As per the reviewers suggestion the statement  “However, the specificity of detailed postmortem inspection was found to be high which could reasonably detect TB-like lesions indicated in table 3  is replaced by ‘However, the specificity of detailed post mortem inspection was found to be high which could maximize the efficiency of lesion detection (table 3)’

  1. Discussion

Q9: Lines 246-248. “The present study was conducted at Debre Birhan municipal abattoir on adult feedlot cattle brought from five different places for slaughter and sold at butcher houses found at Debre Birehan. This can be consumed by the residents and visitors as raw or cooked meat”. Please reorganize these first sentences of the introduction.

 A9:As per the reviewers suggestion the sentences “The present study was conducted at Debre Birhan municipal abattoir on adult feedlot cattle brought from five different places for slaughter and sold at butcher houses found at Debre Birehan. This can be consumed by the residents and visitors as raw or cooked meat”. Were replaced by ‘The present study was conducted at Debre Birhan municipal abattoir that provides slaughter and inspection service to the town. The animals were adult feedlot cattle brought from five different places for slaughter and sold at butcher houses of the town. The butchers were serving the residents and visitors raw or cooked meat’.

Q10:References: Please write the references according to the guidelines of the journal. They must all be changed and not written in alphabetically order.

A10:As per the reviewers suggestion the references are arranged according to the journals guide line.

 Q11:Table 3. +Ve: positive.

A11:As per the reviewers suggestion Ve: positive replaced by +Ve: positive.

Q12: Table 6. The table is not completely visible, please adapt it to the page.

A12: As per the reviewers suggestion the table is adapted to the page and made visible entirely

Q13: Figure 1. Orgion of beef cattle?

A13: Yes, those green areas were the sites from where beef cattle originated.

Reviewer 3 Report

Woldemariyam et al. submitted their actualized manuscript dealing with the Evaluation of Postmortem Inspection Procedures to Diagnose Bovine Tuberculosis. The rewritten manuscript is thoroughly elaborated. The new version was highly improved, better use of English, and I have only a few comments.

  1. Put numbers in square brackets for the used references.
  2. Omit sentence "The abattoir investigation was performed by the first and the second author with a permission of the municipal abattoir." It should be placed in the Authors' Contribution.
  3. Line 166 and line 178: Change to 2.4.1 and 2.4.2.
  4. Line 181: Table 6 before Tables 1,2,3,4,5? Or put Materials and Methods at the end of the manuscript.
  5. Line 192: Omit hyphen between MS and Excel and add a version of the software.6. Again, as I mentioned in the previous peer review, add the information about funding, the author's contributions, and acknowledgment sections.

Author Response

Ms.Ref.No.: animals-1319917

Dear Editor

Please find herewith our point by point correction of the valuable comments of reviewer 3 regarding manuscript entitled “Evaluation of Postmortem Inspection Procedures to Diagnose Bovine Tuberculosis at Debre Birhan Municipal Abattoir; authored by Fanos et al. We revised our manuscript according to the suggestions of the Reviewers. We also included an annotated version of the manuscript with highlighted changes. We responded as follows to the comments:

Sincerely

Jan Paeshuyse

Reviewer  3

Comments and Suggestions for Authors

Woldemariyam et al. submitted their actualized manuscript dealing with the Evaluation of Postmortem Inspection Procedures to Diagnose Bovine Tuberculosis. The rewritten manuscript is thoroughly elaborated. The new version was highly improved, better use of English, and I have only a few comments.

Q1:Put numbers in square brackets for the used references.

A1:As per reviewers suggestion the references were corrected to have square brackets.

Q2:Omit sentence "The abattoir investigation was performed by the first and the second author with a permission of the municipal abattoir." It should be placed in the Authors' Contribution.

A2:As per suggestion of the reviewer this sentence was moved to the author’s contribution ’The abattoir investigation was performed by the first and the second author with a permission of the municipal abattoir’

Q3:Line 166 and line 178: Change to 2.4.1 and 2.4.2.

A3:As per the reviewers suggestion the followings are corrected

Q4:2.6.1. Routine Post Mortem Inspection (RPMI) to 2.4.1. Routine Post Mortem

A4: Inspection (RPMI) and 2.6.2. Detailed Post Mortem Inspection (DPMI) to 2.4.2. Detailed Post Mortem Inspection (DPMI)

Q5:Line 181: Table 6 before Tables 1,2,3,4,5? Or put Materials and Methods at the end of the manuscript.

A5:As per the suggestions the tables are numbered and arranged in order of appearance

Q6:Line 192: Omit hyphen between MS and Excel and add a version of the software.6. Again, as I mentioned in the previous peer review, add the information about funding, the author's contributions, and acknowledgment sections.

A6:As per the reviewers suggestion the hyphen is deleted between MS and Excel. MS-Excel is changed to MS Excel (version: 14.0.4734.1000)

Round 2

Reviewer 1 Report

Simple summary
L 18:  Change “abattoir inspection” to “meat inspection”. Or do the authors refer to the inspection of all facilities?

L 27 : first time TB is used

Abstract
L 40-44: It seems that the routine meat inspection consists in visual inspection, palpation and incisions of carcasses and organs and offal. However, the detailed meat inspection is described as “inspection of each carcasses in detail”. What does it mean “detail”? In material and methods L 171-179 is well explained.

Introduction
L 66: Remove either “bovine tuberculosis” or “bTB”, as it was explained in L 65

L 96-98: Please rewrite the sentence, because is not the genotypic characteristics limited, but the information about the genotypic characteristics.

L 105: bTB has been used several times before, no needed to explain again. Please see my comment below and in previous review rounds regarding consistency of terminology.

Materials and methods
L 110- 114: Please revise the language for a better understanding.

L 123-124: This sentence reads that all animals were older than 4 years old, but Table 5 reads that 134 animals were 3 to 5 years old. Please revise

L 131: bTB has been used several times before, no needed to explain again. Please see my comment below and in previous review rounds regarding consistency of terminology.

L 130-131: this sentence is repetitive of L 116-117.

L 137-138: In previous lines 111-114 it reads 15 week out of 8 months. Please correct

L 148:  is repetitive of L 134-135

Still it is not clear how the authors decided to study 384 animals. Inspection was carried out 4 times per week, with 6 animals inspected per day equals 24 animals per week. If this study lasted 15 weeks, it should be 360 animals inspected. And based on the estimation of sample formula n = (1.96*0.5*(1-0.5))/0.5^2  = 196.

Results
Table 4 is not mentioned in the text. And it is repetitive of the information provided in Table 2.

Table 6: Lesions found in RPMI (lungs/retropharyngeal) is different of the lesion found in DPMI?  Could the authors combine Table 3 and Table 6?

L 210-211: Culture method in laboratory facilities detects/isolates Mycobacterium, and it is usually done in samples taken after detection of lesions. Lesions are detected by the inspection (visual, palpation, incision, etc.).  Culture is not a gold standard for detection of lesion but for isolating/detection of the bacteria. Please correct

L 225-226: Table 6 reports the distribution of lesions in the different tissues and nodes, but not the pathological changes as stated in this sentence.

Discussion
L 257-266: Authors explained that one general reason for the low percentage of detection of lesions is that personnel lack adequate training. However, in this study the inspectors and investigators were veterinarians with special training on meat inspection as stated in L 127-128. So, the lack of training was not the reason for missing 89% of infected carcasses. Also in this study 6 animals were inspected per day, which is not a large number. Could the authors suggest any reason why they missed 89% of infected carcasses?

L 264: see my previous comment in material and methods regarding the age of the animals

L 283-285: It is repetitive of L 267-269. I would strongly recommend to combine L 267-273 and L 283-290?

Other comments
1. Revise the format of references:
In the text, see for example L 64, L 75, L 83, L 90, L 160, L 171, L 243, L 245, L 2417
According to the guidelines for authors (https://www.mdpi.com/journal/animals/instructions): “In the text, reference numbers should be placed in square brackets [ ], and placed before the punctuation; for example [1], [1–3] or [1,3].”

In the list: For example, some have the initials of authors missed [3], some do list all authors [6, 7, 1, 10, 12, 32]. And there are extra dots, commas in many of them

2. Revise the language:
Verb tenses (e.g. L 30-31: change “was” for “were”, L 35: change “was” for “is”),
Sentence construction (e.g. L 38-39, L 43-44, L 274),
Other, for example:  L 23: change “lesion” for “lesions”,  L 24: remove “prevalence”,  L 50 change “majorities” for “majority”, L 124 change “greater” for “older”

3. Be consistent with the terminology, for example bTB (L 81,100, 173, 201) or bovine tuberculosis (L 66, 72, 102, L 292), bovine TB (L 84, L 302), TB (L 125, 252, L 254)

Author Response

Manuscript :animals-1319917

Dear Editor

Please find herewith our point by point correction of the valuable comments of reviewer 1 regarding manuscript entitled “Evaluation of Postmortem Inspection Procedures to Diagnose Bovine Tuberculosis at Debre Birhan Municipal Abattoir ”; authored by Fanos et al. We revised our manuscript according to the suggestions of the reviewers including all the comments. We also included an annotated version of the manuscript with track changes. We responded as follows to the comments:

Sincerely

Jan Paeshuyse

Simple summary

Q1. L18: Change “abattoir inspection” to “meat inspection”. Or do the authors refer to the inspection of all facilities?

A1. As per the reviewer’s suggestion “abattoir inspection” is changed in to “meat inspection”.

Q2. L27: first time TB is used

A2. As per the reviewer’s suggestion: “However, the routine abattoir meat inspection only detected 0.5% of the carcasses examined as having Tuberculosis (TB) lesions is written in full.”

Abstract

Q3. L40-44: It seems that the routine meat inspection consists in visual inspection, palpation and incisions of carcasses and organs and offal. However, the detailed meat inspection is described as “inspection of each carcass in detail”. What does it mean “detail”? In material and methods L171-179 is well explained.

A3. As per reviewer’s suggestion a detailed explanation is written as follows: In this case the seven lobes of the two lungs, lymph nodes and organs were also thoroughly examined. The cut surfaces were examined under a bright light sources for the presence of abscesses, cheesy masses, and tubercles.

Introduction

Q4.L 66: Remove either “bovine tuberculosis” or “bTB”, as it was explained in L 65

A4. As per reviewer’s suggestion we now made the following changes: Bovine tuberculosis is a characteristic contagious disease affecting domestic animals and humans with granulomatous nodule (tubercles) formation in the affected organs.

Q5.L 96-98: Please rewrite the sentence, because is not the genotypic characteristics limited, but the information about the genotypic characteristics.

A5. As per reviewer’s suggestion we rewrote the sentence as follows: Furthermore, as a result of missing lesions at abattoir level, the information about the genotypic characteristics of M. bovis strains affecting the cattle population in Ethiopia is limited.

Q6.L 105: bTB has been used several times before, no needed to explain again. Please see my comment below and in previous review rounds regarding consistency of terminology.

A6. As per reviewer’s suggestion bTB is used throughout the text, except when it is used for the first time in the text.

Materials and methods

Q7.L 110- 114: Please revise the language for a better understanding.

A7. We now revised the sentence as follows: November, December, March and April are religious fasting months for the local orthodox Christian majority of the town of Debre Birhan. During these months no animals are slaughtered and meat consumption is prohibited. Hence, during these months no carcasses could be inspected. Therefore the study took place in October, January, February and May. During every week of each of these months there were 4 days of sampling and during each sampling day 6 animals were inspected. In total there were 64 sampling days with a total of 384 carcasses inspected.

Q8. L123-124: This sentence reads that all animals were older than 4 years old, but Table 5 reads that 134 animals were 3 to 5 years old. Please revise

As per the reviewer’s suggestion

A8.The age of all cattle slaughtered during the study duration was 3 -8 years of age and categorized as adult animals.

Q9.L 131: bTB has been used several times before, no needed to explain again. Please see my comment below and in previous review rounds regarding consistency of terminology.

A9. We now corrected this accordingly.

Q10.L 130-131: this sentence is repetitive of L 116-117.

A10. We now deleted this sentence.

Q11.L 137-138: In previous lines 111-114 it reads 15 week out of 8 months. Please correct

A11. Please, see our answer to question Q7.L 110- 114.

Q12.L 148:  Still it is not clear how the authors decided to study 384 animals. Inspection was carried out 4 times per week, with 6 animals inspected per day equals 24 animals per week. If this study lasted 15 weeks, it should be 360 animals inspected. And based on the estimation of sample formula n = (1.96*0.5*(1-0.5))/0.5^2  = 196.

A12.Dear Reviewer, The calculation is as follows

n = Z2* pexp*(1-pexp) *100

d2

1.962*0.5(1-0.5)/0.52 * 100

                                                                     0.9604/0.25

= 384

Results

Q13.Table 4 is not mentioned in the text. And it is repetitive of the information provided in Table 2.

A13.As per the suggestions of the reviewer Table four is deleted as it is a repetitive of  table two. And the other tables are corrected accordingly. The table number is reduced 5.

The overall lesion prevalence of bTB in cattle slaughtered at Debre Birhan municipality abattoir during the study period was 4.7% (18/384) at 95% CI [2.89, 7.46] based on detailed postmortem inspection as indicated in Table 2, 3 and 5. However, only 0.5% (2/384) at 95% CI [0.09, 2.08] of cattle was found to have detectable TB like lesions by the routine abattoir meat inspection as showed in Table 2.

The sensitivity of routine post mortem inspection at Debre Birhan abattoir in finding TB-like lesions was 11%. From the result of this study, the probability of missing an animal with TB suspected lesion during routine abattoir meat inspection was estimated to be 89%. However, the specificity of detailed post mortem inspection was found to be high which could maximize the efficiency of lesion detection (Table 2).

Q14.Table 6: Lesions found in RPMI (lungs/retropharyngeal) is different of the lesion found in DPMI?  Could the authors combine Table 3 and Table 6?

A14. Table 3 describes the granulomatous lesion identified whereas table 6 describes the distribution of lesion in different nodes and organs. So it is impossible to merge them.

Q15. L 210-211: Culture method in laboratory facilities detects/isolates Mycobacterium, and it is usually done in samples taken after detection of lesions. Lesions are detected by the inspection (visual, palpation, incision, etc.).  Culture is not a gold standard for detection of lesion but for isolating/detection of the bacteria. Please correct

A15. The sentence was deleted

Q16. L 225-226: Table 6 reports the distribution of lesions in the different tissues and nodes, but not the pathological changes as stated in this sentence.

A16. As per suggestions of the reviewer this was corrected and the table numbering is now changed to table 5 as table four was deleted.

This table (table 5) reports the distribution of lesion in different tissues and nodes.

Discussion

Q17.L 257-266: Authors explained that one general reason for the low percentage of detection of lesions is that personnel lack adequate training. However, in this study the inspectors and investigators were veterinarians with special training on meat inspection as stated in L 127-128. So, the lack of training was not the reason for missing 89% of infected carcasses. Also in this study 6 animals were inspected per day, which is not a large number. Could the authors suggest any reason why they missed 89% of infected carcasses?

As per the reviewers concern the sentence and possible explanations are indicated as follows

A17.In addition, lack of interest of inspectors as well as absence of institutional will to incorporate or enforce both methods of examinations as a standard meat inspection workflow. This two might be possible reasons for the inefficiency of the service.

Q18.L 264: see my previous comment in material and methods regarding the age of the animals

As per the reviewers suggestion the age groups are corrected as follows

A18.In all age groups (3-5 and 6-8years) the lesion was identified in lower rate

Q19.L 283-285: It is repetitive of L 267-269. I would strongly recommend to combine L 267-273 and L 283-29 0?

As per the reviewers suggestions the two paragraphs as follows

A19. Our finding implies that large proportion, i.e.89%, of tuberculosis infected carcasses passed undetected and the meat is approved for human consumption. Previous studies have also indicated the probability of 85%, 84% and 70% [24], [26], [34] of missing an animal with TB lesion. Therefore, detailed meat inspection can be considered as a better procedure to detect TB lesions.

 The most probable explanation for the failure of standard meat inspection to correctly detect tuberculosis infection could be the manner of examination [24]. It was noticed that in standard meat inspection procedure only a few sites (organs) are often inspected at a glance due to the work burden of inspecting large number of animals each day as described by Corner, (1994)[24]. In addition, lack of interest of inspectors as well as absence of institutional will to incorporate or enforce both methods of examination as a standard meat inspection workflow. These two could be possible reasons for the inefficiency of the service.

Other comments

Q20.1. Revise the format of references: In the text, see for example L 64, L 75, L 83, L 90, L 160, L 171, L 243, L 245, L 2417 According to the guidelines for authors (https://www.mdpi.com/journal/animals/instructions): “In the text, reference numbers should be placed in square brackets [ ], and placed before the punctuation; for example [1], [1–3] or [1,3].”

A20. We now corrected the references according to the guidelines for authors.

Q21. In the list: For example, some have the initials of authors missed [3], some do list all authors [6, 7, 1, 10, 12, 32]. And there are extra dots, commas in many of them

A21. Corrected accordingly

Q22.2. Revise the language: Verb tenses (e.g. L 30-31: change “was” for “were”, L 35: change “was” for “is”), Sentence construction (e.g. L 38-39, L 43-44, L 274), Other, for example: L 23: change “lesion” for “lesions”, L 24: remove “prevalence”, L 50 change “majorities” for “majority”, L 124 change “greater” for “older”

A22. The suggestions are now corrected.

Q23.3. Be consistent with the terminology, for example bTB (L 81,100, 173, 201) or bovine

A23. As per suggestion we now made the use of bTB or bovine TB consistently throughout the text.

Round 3

Reviewer 1 Report

Thanks for your review. The manuscript has improved considerably since the first time I read it.

This manuscript is a resubmission of an earlier submission. The following is a list of the peer review reports and author responses from that submission.

Round 1

Reviewer 1 Report

In this study, authors investigated the prevalence of bTB lesion and their distribution in different tissues in zebu animals slaughtered at Debre Brihan municipality abattoir. They also compared two methodologies of inspection: routine and detailed post-mortem inspection, and identify potential risk factors associated. They confirm that bTB is present in adult zebu, mainly infected via respiratory route, as lesions were mostly located in lungs and the associated lymph nodes.
As authors highlighted, this type of studies are needed in order to raise the awareness about the consequences of the zoonotic bTB by consuming contaminated meat; as well as the importance of training meat inspectors.

However, the manuscript have some flaws, for example: uncomplete description of the sampling design and the abattoir, results from the routine and detailed antemortem inspections and on the method to select the risk factors for the analysis or why a multivariate analysis was not performed. Limitations of the work should be highlighted in the discussion.
Therefore, I recommend that a revision. I explain my concerns in more detail below, and I ask that the authors to address them.

Major points to improve the article:

Simply summary
According to the instructions for authors “it contains a clear statement of the problem addressed, the aims and objectives, pertinent results, conclusions from the study and how they will be valuable to society”. The current summary does mention the problem, which it can be guessed (fail in detecting bTB at the abattoir). Authors’ main aim was to compare the two procedures, assess the lesion distributions in different tissues and organs, and maybe to establish associations between prevalence of bTB lesion and some risk factors (e.g. sex, age, BCS). I would recommend to review/rewrite the summary

Abstract
According to the Instructions for authors, the abstract should include a background where the research question and the purpose of the study should be mentioned. Aim is described as “to estimate tuberculosis (TB) suspected lesion prevalence of bovine tuberculosis (bTB) and evaluate the efficiency of meat inspection procedures to detect carcass infected with Mycobacterium bovis”, however, in the introduction is defined as “specific objectives of estimating the abattoir lesion prevalence of bTB and assessing the lesion distributions in different tissues and organs in animals slaughtered at Debre Birhan municipality abattoir”. Results summarize the main findings, and other not included in the aim: the existing associations between lesion of bTB prevalence and potential risk factors. I would recommend revise the abstract.

L33-34: How the authors define “relatively moderate”? Readers can be confused as many interpretations are possible, thus I would recommend to clarify.

Introduction
Introduction presents a detailed background about the importance of human and bovine TB, as well as other studies on animal and herd prevalence. Very briefly reports about other studies carried out to detect bTB in abattoirs in Ethiopia, describing why they are not representative of country. Is this lack of representativeness the gap of knowledge that this study wants to fill up? In such case, I would recommend to highlight it. I would also recommend to outline the research question of this study or justify why this study was carried out. What is the difference compared to the previous articles? Are now better diagnostic facilities, or better surveillance plans designed, etc..?

L74-75: The sentence is confusing. Is consuming animal products raw a risk factor for bovine TB?, Do zebus in Ethiopia eat animal products raw? Or do they refer to human TB?. Please, revise. In addition, are there only 2 risk factors of bTB in Ethiopia?

Materials and methods
Inspection carried out 4 times (or days?) per week, with 6 animals inspected per day equals 24 animals per week. If this study lasted 8 months = 32 weeks, it should be 768 animals inspected. Can authors explain how they decided to study 384 animals?

How the 6 animals were selected out of the average 38 animals slaughtered per day, i.e. what was the selection criteria? In addition, the formula provided to calculate the sample size requires an expected prevalence, which one do the authors use?

Who carry out the study (i.e. personal in the abattoir or the authors/investigators)? In addition, more information about the abattoir should be included, for example: Does the abattoir only slaughter zebu? Is the personal in the abattoir vet or meat inspector? This information would provide also a basis for discussion.

L89: Authors indicated “most likely originated”, what does it mean (e.g. all of them, 75% of animals, etc.) Please clarify.

Results and discussion
No results of the antemortem inspections are presented, although its details are presented in L86-87: “Antemortem and postmortem procedures were performed in both routine and detailed meat inspections” and in L115-124, which describe how the antemortem inspections were carried out. Thus, authors collected information about other potential risk factors, but it is not mention anymore. Some additional information is needed, for example, why and how have these potential risk factors been discarded?

Do the authors carry out a multivariate analysis to evaluate collinearity (L189) as mentioned? In such case, please provide the results.

L184-185: How authors infer that DPMI has a sensitivity of 100%? They have not compared to DPMI with the considered Ethiopian gold standard Mycobacterium culture (L59-60). Please revise.

L 223: Authors stated that cattle was brought from five different places, while previously was “most likely originated” (L89). Please clarify (see my previous comment).

L229-239: Authors cited for the first time some other articles with similar results. At least some of these articles should be cited also in the introduction as a background for this study, in order to explain why this study was needed or how this study is different from the others. In addition for the discussion, which are the differences of those studies with the current study? I mean, endemicity of the disease and high infection rate prevailing in the general population of slaughtered cattle can be inferred from the other studies.

L241-255: These two paragraphs discuss about the same, the probability of missing animals with bTB lesion. I would suggest to combine them in only one.

L251: Depending on the abattoir, region, country, etc. 38 animals as average per day, is not consider a large number. Could authors comment on this?

L253-255: Have the employees of the abattoir lack of competence in meat inspection? In such case, it should be mention in materials and methods (survey of employees) and in results. (See previous comment about providing more information on the abattoir).

L263: Do authors intend to say that the prevalence between age groups was not different? Please correct if such is the case.

L272-277: Is this conclusion specific for old cattle? In L46-49 the respiratory route of infection is the commonest in old cattle. Please clarify.

Tables and Figures
L 172: only Table 1 describes RPMI and DPMI.

Is there a missing row in Table 3?

I would recommend to join table 1 and table 3, as some of the results presented are repetitive.

Table 2 is not mentioned in the text. Also, table 2 indicates that there were 18 Granulomatous lesion found during the detailed postmortem inspection, however Table 5 reads that there were 36. It is confused, maybe the authors could combine these both tables (2 and 5).

L172-175: does not refer to Table 4, which is the risk factors associations table.

Figure 2: Red and clear blue arrow are not shown in the photos.

References
11, 21, 31: they are incomplete.

11 and 12: full names are provided while in other only initials.

L167: Reference 27 is not appropriate. Reference 27 indicates which software was used, it is not a reference.

Minor points and other inconsistencies:

L28-29: What does this sentence mean: “The current finding necessitates a sound epidemiological safety measures to ensure safety of the consumers”?

L84: Times a week means days per week?

Authors use bTB most of the times, but also bovine TB and bovine tuberculosis. For clarity purpose, be consistent and use the same form along the manuscript.

Reviewer 2 Report

It is an interesting paper, but this article needs a lot of improvements to be suitable for publication in Animals. First of all, there is the need to improve and re-write some sentences because they are unclear and difficult to understand. I suggest modifying the tables and adding another one (see specific comments). The authors must adapt the manuscript according to the guidelines of the journal (the references are incorrectly written both in the text and at the end of the manuscript). Moreover, I suggest deleting some of these references, replacing them with other updated manuscripts. The “Discussion” needs to be improved while the “Conclusion and Recommendations” are well written. In my view, this manuscript is suitable for publication in Animals after Major Revisions to improve the document.

Specific comments

In my opinion, some information regarding the affiliations of the authors must be added such as the address and their initials, necessary for the “Authors contributions”, according to the guidelines of the journal.

Simple summary

Line 12 and throughout the manuscript. Little space between summary: and Tuberculosis. Please check the whole manuscript for this misspelling. Please write the word tuberculosis in the proper way. Moreover, write TB after Tuberculosis since it appears here for the first time.

Line 12. “It can be transmitted”.

Line 15 and throughout the manuscript. There are two spaces between detailed and meat inspections. Please check the whole manuscript for this misspelling.

Abstract

Line 21. Please correct the sentence as follows: “from October 2016 to May 2017 to estimate the prevalence of suspected lesions of bovine tuberculosis (bTB)”.

Line 29. Please delete one dot.

Line 31. “gastro-intestinal” or “gastrointestinal” tract.

Keywords: I suggest delete Debre Birhan and add another term such as lesions or other.

  1. Introduction

Line 43. Please write bTB first in the whole form and then in its acronym because it appears here for the first time in the text.

Line 43. Please add “This is a characteristic” or “It consists of a characteristic”.

Lines 47-49. Please re-write the sentence because it is difficult to understand.

Line 50. “Animals affected by bovine tuberculosis” lose...

Line 56. ...and “because of the susceptibility” of AIDS...

Line 59. “although” Mycobacterium...

Line 60. Please delete one dot.

Lines 63-64 and throughout the manuscript. Please add the name of the reference or write the numbers at the end of the sentence. It is not correct writing: “Among these, [11] reported 1.5% and 7.4% herd and animal level prevalence whereas [12] reported herd level prevalence of 17-29% and animal level prevalence of 5.2% at the cut-off >4 mm”. Author et al. [11]. Please check the whole manuscript for this mistake.

Lines 67-69. Please re-write the sentence because it is unclear.

Line 71. Move the comma after bovis and delete it after “in particular”.

  1. Materials and Methods

Line 84. “at the abattoir”.

Line 85. “per day” during the study period. All of them were...

Line 89. Delete the comma after that.

Line 90. (Figure 1). “The majority were male animals”.

Figure 1. The figure is difficult to read, the writings must necessarily be larger.

Line 163. Replace “comprises” with a more scientific term and write it in the past tense.

Line 105. Delete the sentence: “In total, the carcasses of 384 heads of cattle were inspected” because it is already written above.

Lines 119-121, 128-129, 139. See “Lines 63-64 and throughout the manuscript”. Please do not use too many times “according to” or other similar expressions but write only the reference number at the end of the sentences to make the manuscript smoother.

Line 147. “prescapular”.

Lines 141-150. In my opinion it would be easier to add a table in order to better highlight the seven lobes of the lungs, lymph nodes and organs and not to write all this information in the text because it is boring and difficult to memorize. This is an example:

Lobes of the lungs

Left apical, left cardiac...

Lymph nodes

Parotid, submaxillary...

Organs

Liver, kidneys...

Lines 166-167. Delete “but insignificant when P> 0.05”.

  1. Results

Line 174. “as showed in Table 4”.

Tables 1 and 3. I believe that Tables 1 and 3 were modified during the submission process because the exact arrangement of the lines and of the columns is difficult to understand. Moreover, in the Table 3 what are: “+ Ve” and “-Ve”? What about: “Se”, “Sp”, “PV+” and “PV-”? Please add a caption below the Table 3.

Table 4. The number 384 could only be written once at the level of N and not three times. Moreover, you can remove the first P-value column because it is empty. I would suggest organizing the table to let it enter the sheet vertically and not horizontally, for easier reading.

Line 206. “are presented in...”

Table 5. “Tissue and Lymph nodes (LN)”. Please delete Lymph nodes (LN) after Mediastinal as well as the other LN in the second column because LN is present in the title.

Line 218. Please delete the second dot after abattoir.

Figure 2. I think the images overlap. I don’t see the red and light blue arrows. Please change the layout of the figures.

Lines 219-220. The arrows have shifted and covered the words. Please correct.

  1. Discussion

Lines 229-238. See “Lines 63-64 and throughout the manuscript”. Please do not use too many times “the reports of” or “by” or “from” but write only the reference number at the end of the sentences to make the manuscript smoother.

Lines 229-231. “%” instead of percent; “prevalence”. Please re-write the entire sentence because it is unclear.

Line 232. Add a comma after However.

Line 236. “%” instead of percent; and comma before respectively.

Line 238. Delete “which is much higher”.

Line 241. The present study revealed “that” the probability...

Line 243. Please add the dot before Therefore.

Line 252. et al. [26].

Line 258. 13.2%.

Line 264. [28,39].

Line 267-269. Please modify the sentence because it is unclear.

  1. Conclusion and Recommendations

Lines 284-288. Please modify the sentences as follows: “Based on the above concluding remark the following recommendations are made: (i) training of the meat inspectors working at Debre Birhan municipal abattoir so as to safeguard more people...or to protect as many people as possible; (ii) an in-depth epidemiological study focusing on culturing, isolation and identification of the agent as well as molecular characterization of the strains cycling in the study area; (iii) public...”

References: Please write the references according to the guidelines of the journal. They must all be changed.

Reviewer 3 Report

The manuscript of Fanos Tadesse Woldemariyam et al. titled "Evaluation of Postmortem Inspection Procedures to Diagnose Bovine Tuberculosis at Debre Birhan Municipal Abattoir" is devoted to the study of postmortem inspection of 384 heads of cattle in Ethiopia to diagnose bovine tuberculosis. The authors used antemortem and postmortem inspection methods to estimate tuberculosis suspected lesion prevalence.
Since the manuscript was written by non-English-speaking authors, I would highly recommend significant professional editing of the English. The translation into English is extremely inaccurate and difficult to read because of misspellings. Authors used singular instead of plural, and vice versa. For example: under cooked should be the one word "undercooked" (line 13), "shown" instead of "showen" (line 62), "tables" instead of "table" (line 172), "abattoir" instead of "abattior" (in some places of the manuscript), "abattoirs" instead of "abattior" (line 233), and so on.
Although the manuscript has some new data (e.g. the only one major conclusion that the prevalence was increased from 0,52% to 4.7% based on detailed postmortem inspection), this is not enough to devote one regular article to this observation. I suggest to re-write the results as more concise short communications.
The methods used are highly subjective and not described in detail. I suggest briefly describe measurements of BCS (body condition scores) in the text since most readers won't look at reference 21.
Another weakness of the study is a lack of bacteriological data, since the authors mentioned that tuberculosis was caused by Mycobacterium bovis in all cases, they did not show any data about it.

The minor issues and comments are:
Line 36: What is the reason to write here "Debre Birhan public abattoir" instead of "Debre Birhan municipal abattoir", as elsewhere in the manuscript? Which one is the correct name for the slaughterhouse?
Line 37: Could you replace the keyword "Debre Birhan and Meat inspection", since it is very unlikely that someone will use it for search?
Line 59: Italicize "Mycobacterium".
Line 60: "." repeated two times.
Lines 89-90: Why the geographical locations are italicized here?
Line 111: What is "x" in the formula?
Line 139: Reference 24 goes after reference 25 (line 135)?
Line 160: Provide names of companies and countries for the software.
Line 186: Table 3 is not explained in the text of the manuscript. What is Se, Sp, PV+, PV-? Also, I suppose that there is a problem with formatting.
Line 194: What is the difference between -0.00 and 0.00 in the Chi-square column data?
Line 195: "BSC" replace with "BCS".
Line 207: Are the DPMI lesions not included RPMI lesions? Are they disappeared on this stage? Why?
Line 207: Total % is not 100%, but 100.23%!
Line 207: S/No is not a sample number, but an organ number, I suppose.
Lines 211-212: Total % for the lesions is 100.1%!
Line 219: I can't see arrows (red and olive) in the upper pictures.
Line 224: Replace "Debrebirehan" with "Debre Birehan".
Line 272: Where is the reference for Corner (1994)?
Reference [1] and [33]: Is this one or two references here?
Reference [1] and so on: Please, use the correct titles for the journals everywhere in the references!
References [11] and [12]: provide journal names, numbers, pages.
References [13], [18], [33]: remove a's and b's from years.
References [21], [23], [31], [32], [35]: number of pages?
References [32], [35], and [36]: I suspect, that it's impossible to reference theses.

Moreover, I can't find information about funding, the author's contributions, and acknowledgment sections.

Also, I did not found mention of references [16] and [37] in the text of the manuscript.